# Towards Black-Box Membership Inference Attack for Diffusion Models

**Jingwei Li** [1 2]   **Jing Dong** [3]   **Tianxing He** [1 2]   **Jingzhao Zhang** [1 2]

## Abstract

Given the rising popularity of AI-generated art and the associated copyright concerns, identifying whether an artwork was used to train a diffusion model is an important research topic. The work approaches this problem from the membership inference attack (MIA) perspective. We first identify the limitation of applying existing MIA methods for proprietary diffusion models: the required access of internal U-nets. To address the above problem, we introduce a novel membership inference attack method that uses only the image-to-image variation API and operates without access to the model's internal U-net. Our method is based on the intuition that the model can more easily obtain an unbiased noise prediction estimate for images from the training set. By applying the API multiple times to the target image, averaging the outputs, and comparing the result to the original image, our approach can classify whether a sample was part of the training set. We validate our method using DDIM and Stable Diffusion setups and further extend both our approach and existing algorithms to the Diffusion Transformer architecture. Our experimental results consistently outperform previous methods.

## 1. Introduction

Recently, there has been a surge in the popularity of generative models, with diffusion models in particular, gaining huge attention within the AI community (Sohl-Dickstein et al., 2015; Song & Ermon, 2019; Song et al., 2020b). These models have demonstrated remarkable capabilities across various tasks, including unconditional image generation (Ho et al., 2020; Song et al., 2020a), text-to-image

generation (Rombach et al., 2022; Yu et al., 2022; Nichol et al., 2021) and image-to-image generation (Saharia et al., 2022a). This surge has given rise to powerful AI art models such as DALL-E 2 (Ramesh et al., 2022), Stable Diffusion (Rombach et al., 2022), and Imagen (Saharia et al., 2022b). AI-generated art has a promising future and is expected to have widespread impact.

Effective training of diffusion models requires high-quality data. It is thus crucial to design an algorithm that can identify whether a specific artwork has been used during the training of a model, thereby providing protection for these artworks and detecting misuse of data. This is especially important due to the rapid growth of generative models, which has raised concerns over intellectual property (IP) rights, data privacy, and the ethical implications of training on copyrighted or proprietary content without consent. As these models are increasingly deployed across industries, detecting whether a specific piece of content was used in training can help prevent unauthorized use of artistic works, protecting creators' copyrights and ownership rights. This is a classic problem in the field of machine learning, first introduced by Shokri et al. (2017) and named "membership inference attack".

A series of studies have been conducted on membership inference attacks against diffusion models. Hu & Pang (2023) was the first to examine this issue, utilizing the loss function values of diffusion models to determine whether an image is in the training set. Duan et al. (2023) and Kong et al. (2023) extended this work by relaxing assumptions about the access requirements of the model.

Despite great progress, previous methods are not yet ready for MIA in proprietary diffusion models. Most existing approaches heavily rely on checking whether the U-net of the model predicts noise accurately, which is not practical since most commercial diffusion models available today offer only API access, while the U-net remains hidden.

To address the above issue, we propose a membership inference attack method that relies on the variation API and does not require access to the denoise model (e.g., U-net). We observe that if we alter an image using the target diffusion model's variation API, the sampling process will be captured by "a region of attraction" if the model has seen this image during training (illustrated in Figure 2). Based on the

[1]Institute for Interdisciplinary Information Sciences, Tsinghua University [2]Shanghai Qizhi Institute [3]The Chinese University of Hong Kong. Correspondence to: Jingwei Li <ljw22@mails.tsinghua.edu.cn>, Jingzhao Zhang <jingzhaoz@mail.tsinghua.edu.cn>.

*Proceedings of the $42^{nd}$ International Conference on Machine Learning*, Vancouver, Canada. PMLR 267, 2025. Copyright 2025 by the author(s).

above observation, we propose the REDIFFUSE algorithm for MIA with image-to-image variation API, and can detect member images without accessing the denoise model. Our main contributions are listed as follows:

1. We propose a membership inference attack method that **does not require access to the model's internal structure**. Our method only involves using the model's variation API to alter an image and compare it with the original one. We name our method REDIFFUSE.

2. We evaluate our method using DDIM (Song et al., 2020a) and Stable Diffusion (Rombach et al., 2022) models on classical datasets, including CIFAR10/100 (Krizhevsky et al., 2009), STL10-Unlabeled (Coates et al., 2011), LAION-5B (Schuhmann et al., 2022), etc. Our method outperforms the previous methods.

3. We extend both existing algorithms and our own algorithm to the Diffusion Transformer (Peebles & Xie, 2023) architecture, implementing the membership inference attack within this model framework **for the first time**. Experimental results demonstrate that our algorithm is consistently effective.

## 2. Related Works

**Diffusion Model**    The diffusion model, initially proposed by Sohl-Dickstein et al. (2015), has achieved remarkable results in producing high-quality samples across a variety of domains. This ranges from image generation (Song & Ermon, 2019; Song et al., 2020b; Dhariwal & Nichol, 2021), audio synthesis (Popov et al., 2021; Kong et al., 2020; Huang et al., 2022), and video generation (Ho et al., 2022a;b; Wu et al., 2023), etc. Among existing diffusion models, the Denoising Diffusion Probabilistic Model (DDPM) (Ho et al., 2020) is one of the most frequently adopted. This approach introduces a dual-phase process for image generation: initially, a forward process gradually transforms training data into pure noise, followed by a reverse process that meticulously reconstructs the original data from this noise. Building on this model, there have been numerous follow-up studies, such as Stable Diffusion (Rombach et al., 2022), which compresses images into a latent space and generates images based on text, and the Denoising Diffusion Implicit Models (DDIM) (Song et al., 2020a), which removes Gaussian randomness to accelerate the sampling generation process. These advancements demonstrate the versatility and potential of diffusion models.

**Data Safety and Membership Inference Attack**    In the era of big data, preserving data privacy is paramount. The training of diffusion models may involve sensitive datasets like artists' artworks, which are protected by copyright

laws. Membership inference attacks, initially introduced by Shokri et al. (2017), serve as an effective means to detect potential misuse of data without proper authorization. Its objective is to ascertain whether a particular data sample participated in the training phase of a target model. This approach is instrumental in probing privacy breaches and identifying illicit data utilization. Researchers primarily focus on membership inference attacks for classification models (Salem et al., 2018; Yeom et al., 2018; Long et al., 2018; Li & Zhang, 2021), embedding models (Song & Raghunathan, 2020; Duddu et al., 2020; Mahloujifar et al., 2021), and generative models (Hayes et al., 2017; Hilprecht et al., 2019; Chen et al., 2020).

In the domain of membership inference attacks against diffusion models, Wu et al. (2022); Hu & Pang (2023) use a white-box approach, which assumes access to the entire diffusion model and utilizes loss and likelihood to determine whether a sample is in the training set. Duan et al. (2023); Kong et al. (2023); Tang et al. (2023) have relaxed these requirements, eliminating the need for the entire model. They leverage the insight that samples within the training set yield more accurate noise predictions, thereby achieving high accuracy in membership inference attack tasks. However, they also require the outputs of the U-net, as it is necessary to obtain the noise predictions of intermediate steps. Recently, Pang & Wang (2023) proposed a black-box membership inference attack method against diffusion models. Their method identifies whether a specific image is in a finetuning dataset of 100 images by calculating the difference between the generated image and the target image, using the corresponding prompt as input. Their approach leverages the model's tendency to memorize finetuning images and generate similar outputs. In contrast, we focus on detecting whether an image is in the pretraining dataset, where a pretrained model often produces diverse outputs for the same prompt, making detection more challenging.

## 3. Preliminary

In this section, we begin by introducing the notations used for several popular diffusion models. We first introduce the Denoising Diffusion Probabilistic Model (DDPM) (Ho et al., 2020). Then, we extend to the Denoising Diffusion Implicit Model (DDIM) (Song et al., 2020a) and Stable Diffusion (Rombach et al., 2022), which are variants of DDPM used to accelerate image generation or generate images grounded in text descriptions. Lastly, we discuss Diffusion Transformer (Peebles & Xie, 2023), a model that replaces the U-net architecture with a transformer and achieves higher-quality image generation.

**Denoising Diffusion Probabilistic Model (DDPM)**    A diffusion model provides a stochastic path between an image and noise. The forward process (denoted as $q$) iteratively incorporates Gaussian noise into an image, while the reverse

process (denoted as $p_\theta$) gradually reconstructs the image from noise.

$$q(x_t \mid x_{t-1}) = \mathcal{N}\left(x_t; \sqrt{1 - \beta_t} x_{t-1}, \beta_t \mathbf{I}\right),$$

$$p_\theta(x_{t-1} \mid x_t) = \mathcal{N}\left(x_{t-1}; \mu_\theta(x_t, t), \Sigma_\theta(x_t, t)\right),$$

where $\mu_\theta(\cdot)$ and $\Sigma_\theta(\cdot)$ are the mean and covariance of the denoised image parameterized by the model parameters $\theta$, and $\beta_t$ is a noise schedule that controls the amount of noise added at each step.

**Denoising Diffusion Implicit Model (DDIM)** DDIM modifies the sampling process to improve efficiency while maintaining high-quality image generation. Unlike DDPM, which requires a large number of denoising steps, DDIM uses a non-Markovian process to accelerate sampling.

$$x_{t-1} = \phi_\theta(x_t, t) = \sqrt{\bar{\alpha}_{t-1}} \left( \frac{x_t - \sqrt{1 - \bar{\alpha}_t} \epsilon_\theta(x_t, t)}{\sqrt{\bar{\alpha}_t}} \right)$$
$$+ \sqrt{1 - \bar{\alpha}_{t-1}} \epsilon_\theta(x_t, t),$$

where $\bar{\alpha}_t = \prod_{k=0}^{t} \alpha_k$, $\alpha_t + \beta_t = 1$ and $\epsilon_\theta(x_t, t)$ is the noise predicted by the model at step $t$. This formulation requires fewer sampling steps without compromising the quality of the generated images.

**Stable Diffusion** Stable Diffusion leverages a variational autoencoder (VAE) (Kingma & Welling, 2013) to encode images into a latent space and perform diffusion in this compressed space. The model uses a text encoder to guide the diffusion process, enabling text-to-image generation:

$$z_{t-1} \sim p_\theta(z_{t-1} \mid z_t, \tau_\theta(y)), \quad x = \text{Decoder}(z_0),$$

where $x$ represents the output image, $z_t$ represents the latent variable at step $t$, and the text conditioning $\tau_\theta(y)$ is incorporated into the denoising process to generate the image. This approach significantly reduces computational costs and allows for high-quality image synthesis from textual descriptions.

**Diffusion Transformer** Diffusion Transformer leverages the Vision Transformer (Dosovitskiy, 2020) structure to replace the U-net architecture traditionally used in diffusion models for noise prediction. Its training and sampling methods remain consistent with DDIM, with the only difference being the replacement of noise prediction network $\epsilon_\theta$ with $\epsilon_{\tilde{\theta}}$, where $\tilde{\theta}$ represents a Vision Transformer-based architecture. This approach further enhances the generation quality and ensured that the model possesses good scalability properties.

# 4. Algorithm Design

In this section, we introduce our algorithm. We begin by discussing the definition of variation API and the limitations

of previous membership inference attack methods. In our formulations, we assume DDIM as our target model. The formulations for DDPM are highly similar and we omit it for brevity. We will discuss the generalization to the latent diffusion model in Section 4.3.

## 4.1. The variation API for Diffusion Models

Most previous works on membership inference attacks against diffusion models aim to prevent data leakage and hence rely on thresholding the model's training loss. For instance, Hu & Pang (2023) involves a direct comparison of image losses, while Duan et al. (2023); Kong et al. (2023) evaluates the accuracy of the model's noise prediction at initial or intermediate steps. However, the required access to the model's internal U-net structure prevents applications from copyright protection because most servers typically provide only black-box API access.

In contrast, our method represents a step towards black-box MIA, as we do not directly access the model's internal structure. Instead, we rely solely on the variation API, which takes an input image and returns the corresponding output image. Below, we formalize the definition of the variation API used in our algorithm.

**Definition 4.1** (The variation API). We define the variation API $V_\theta(x, t)$ of a model as follows. Suppose we have an input image $x$, and the diffusion step of the API is $t$. The variation API randomly adds $t$-step Gaussian noise $\epsilon \sim \mathcal{N}(0, I)$ to the image and denoises it using the DDIM sampling process $\phi_\theta(x_t, t)$, returning the reconstructed image $V_\theta(x, t)$. The details are as follows:

$$x_t = \sqrt{\bar{\alpha}_t} x + \sqrt{1 - \bar{\alpha}_t} \epsilon,$$
$$V_\theta(x, t) = \Phi_\theta(x_t, 0) = \phi_\theta(\cdots \phi_\theta(\phi_\theta(x_t, t), t - 1), 0).$$

This definition aligns with the image-to-image generation method of the diffusion model, making access to this API practical in many setups (Lugmayr et al., 2022; Saharia et al., 2022a; Wu & De la Torre, 2023). Some APIs provide the user with a choice of $t$, while others do not and use the default parameter. We will discuss the influence of different diffusion steps in Section 5.5, showing that the attack performances are relatively stable and not sensitive to the selection of $t$. We also note that for the target model, we can substitute $\phi_\theta(x_t, 0)$ with other sampling methods, such as the Euler-Maruyama Method (Mao, 2015) or Variational Diffusion Models (Kingma et al., 2021).

## 4.2. Algorithm

In this section, we present the intuition of our algorithm. We denote $\| \cdot \|$ as the $L_2$ operator norm of a vector and $\mathcal{T} = \{1, 2, \ldots, T\}$ as the set of diffusion steps. The key insight is derived from the training loss function of a fixed

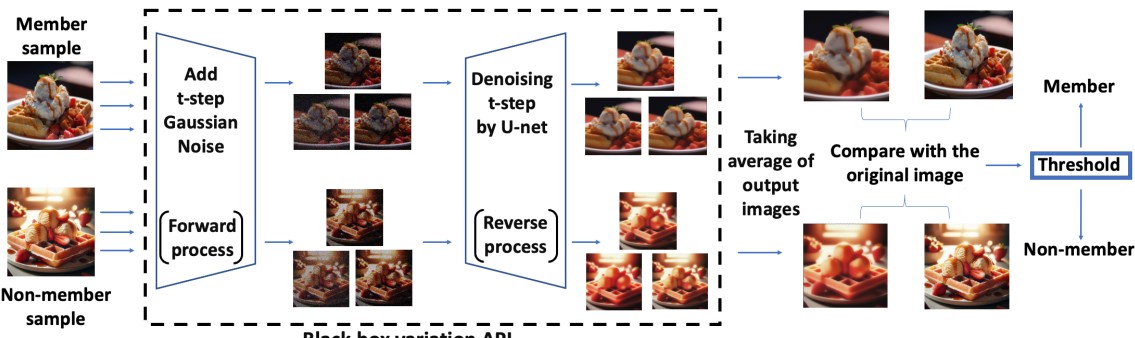

Figure 1: **The overview of REDIFFUSE**. We independently input the image to the variation API $n$ times with diffusion step $t$. We take the average of the output images and compare them with the original ones. If the difference is below a certain threshold, we determine that the image is in the training set.

sample $x_0$ and a time step $t \in \mathcal{T}$:

$$L(\theta) = \mathbb{E}_{\epsilon \sim \mathcal{N}(0, \mathbf{I})} \left[ \left\| \epsilon - \epsilon_\theta \left( \sqrt{\bar{\alpha}_t} x_0 + \sqrt{1 - \bar{\alpha}_t} \epsilon, t \right) \right\|^2 \right] .$$

Denote $x_t = \sqrt{\bar{\alpha}_t} x_0 + \sqrt{1 - \bar{\alpha}_t} \epsilon$, we assume that the denoise model is expressive enough (the neural network's dimensionality significantly exceeds that of the image data) such that for the input $x_0 \in \mathbb{R}^d$ and time step $t \in \mathcal{T}$, the Jacobian matrix $\nabla_\theta \epsilon_\theta(x_t, t)$ is full rank ($\geq d$). This suggests that the model can adjust the predicted noise $\epsilon_\theta(x_t, t)$ locally in any direction. Then for a well trained model, we would have $\nabla_\theta L(\theta) = 0$. This implies

$$\implies \nabla_\theta \epsilon_\theta(x_t, t)^T \mathbb{E}_{\epsilon \sim \mathcal{N}(0, \mathbf{I})} \left[ \epsilon - \epsilon_\theta(x_t, t) \right] = 0 ,$$
$$\implies \mathbb{E}_{\epsilon \sim \mathcal{N}(0, \mathbf{I})} \left[ \epsilon - \epsilon_\theta(x_t, t) \right] = 0 .$$

This is intuitive because if the neural network noise prediction exhibits a high bias, the network can adjust to fit the bias term, further reducing the training loss.

Therefore, for images in the training set, we expect the network to provide an unbiased noise prediction. Since the noise prediction is typically inaccessible in practical applications, we use the reconstructed sample $\hat{x}$ as a proxy. By leveraging the unbiasedness of noise prediction, we show that averaging over multiple independent reconstructed samples $\hat{x}_i$ can significantly reduce estimation error (see Theorem 4.2). However, for images not in the training set, the neural network may not provide an unbiased prediction at these points. The intuition is illustrated in Figure 2.

With the above intuition, we introduce the details of our algorithm. We independently apply the variation API $n$ times with our target image $x$ as input, average the output images, and then compare the average result $\hat{x}$ with the original image. We will discuss the impact of the averaging number $n$ in Section 5.5. We then evaluate the difference between the images using an indicator function:

$$f(x) = \mathbf{1} \left[ D(x, \hat{x}) < \tau \right] .$$

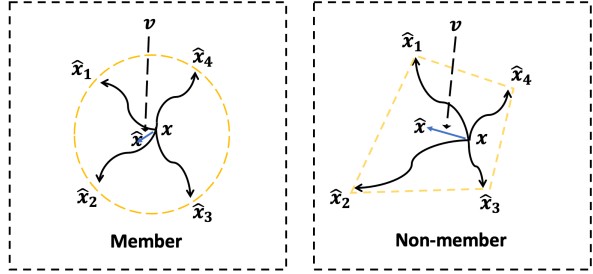

Figure 2: **The intuition of our algorithm design**. We denote $x$ as the target image, $\hat{x}_i$ as the $i$-th image generated by the variation API, and $\hat{x}$ as the average image of them. For member image $x$, the difference $v = x - \hat{x}$ will be smaller after averaging due to $x_i$ being an unbiased estimator.

Our algorithm classifies a sample as being in the training set if $D(x, \hat{x})$ is smaller than a threshold $\tau$, where $D(x, \hat{x})$ represents the difference between the two images. It can be calculated using traditional functions, such as the SSIM metric (Wang et al., 2004). Alternatively, we can train a neural network as a proxy. In Section 5, we will introduce the details of $D(x, \hat{x})$ used in our experiment.

Our algorithm is outlined in Algorithm 1, and we name it REDIFFUSE. The key ideas of our algorithm are illustrated in Figure 1, and we also provide some theoretical analysis in Theorem 4.2 to support it.

**Analysis** We give a short analysis to justify why averaging over $n$ samples in REDIFFUSE can reduce the prediction error for training data. We have the following theorem showing that if we use the variation API to input a member $x \sim D_{\text{training}}$, then the error $\|\hat{x} - x\|$ from our method will be small with high probability.

**Theorem 4.2.** *Suppose the DDIM model can learn a parameter $\theta$ such that, for any $x \sim D_{training}$ with dimension $d$, the prediction error $\epsilon - \epsilon_\theta(\sqrt{\bar{\alpha}_t} x + \sqrt{1 - \bar{\alpha}_t} \epsilon, t)$ is a ran-*

Table 1: Comparison of different methods on four datasets for the DDIM model. Previous methods require access to the U-Net, whereas our methods do not. We use AUC, ASR, and TP as the metrics, TP refers to the True Positive Rate when the False Positive Rate is 1%.

| Method | | CIFAR10 | | | CIFAR100 | | | STL10 | | |
|---|---|---|---|---|---|---|---|---|---|---|
| Algorithm | U-Net | AUC | ASR | TP | AUC | ASR | TP | AUC | ASR | TP |
| Loss (Matsumoto et al., 2023) | □ | 0.88 | 0.82 | 14.2 | 0.92 | 0.84 | 20.9 | 0.89 | 0.82 | 15.6 |
| SecMI (Duan et al., 2023) | □ | 0.95 | 0.90 | 40.7 | 0.96 | 0.90 | 44.9 | 0.94 | 0.88 | 26.9 |
| PIA (Kong et al., 2023) | □ | 0.95 | 0.89 | 48.7 | 0.96 | 0.90 | 47.0 | 0.94 | 0.87 | 29.8 |
| PIAN (Kong et al., 2023) | □ | 0.95 | 0.89 | **50.4** | 0.91 | 0.85 | 39.2 | 0.92 | 0.86 | 28.5 |
| **REDIFFUSE** | ■ | **0.96** | **0.91** | 40.7 | **0.98** | **0.93** | **48.2** | **0.96** | **0.90** | **31.9** |

□: Require the access of U-Net. ■: Do not require the access of U-Net.

## Algorithm 1 MIA through REDIFFUSE

**Input:** Target image $x$, diffusion step $t$, average number $n$, threshold $\tau$, the variation API of the target model $V_\theta$, distance function $D$.

**for** $k = 1, \ldots, n$ **do**

Use the variation API $V_\theta$ to generate the variation image $\hat{x}_k = V_\theta(x, t)$ according to Definition 4.1.

**end for**

Average the reconstructed images from each iteration $\hat{x} = \frac{1}{n}(\hat{x}_1 + \hat{x}_2 + \ldots + \hat{x}_n)$.

**return** "YES" if the distance between the two images $D(x, \hat{x})$ is less than $\tau$, otherwise "NO".

---

*dom variable $X = (X_1, X_2, \ldots, X_d)$ with zero expectation and finite cumulant-generating function for each coordinate (Durret, 2010). Suppose the sampling interval $k$ is equal to the variation API diffusion step $t$. Let $\hat{x}$ be the average of $n$ output images of $x$ using the variation API. Then we have*

$$\mathbb{P}(\|\hat{x} - x\| \geq \beta) \leq d \exp\left(-n \min_i \Psi^*_{X_i}\left(\frac{\beta\sqrt{\bar{\alpha}_t}}{\sqrt{d(1 - \bar{\alpha}_t)}}\right)\right),$$

*where $\beta > 0$ and $\Psi^*_{X_i}$ is the Fenchel-Legendre dual of the cumulant-generating function $\Psi_{X_i}$.*

The theorem suggests that averaging the random variational data from a training set will result in a lower relative error with high probability when we use a large $n$. We defer the proof of this theorem to Appendix C.

### 4.3. MIA on Other Diffusion Models

In this section, we discuss how we can generalize our algorithm to other diffusion models. We note that the variation API for stable diffusion is different from DDIM, as it includes the encoder-decoder process. Again we denote

$\hat{x} = V_\theta(x, t)$, and the details are as follows:

$$z = \text{Encoder}(x), \quad z_t = \sqrt{\bar{\alpha}_t}z + \sqrt{1 - \bar{\alpha}_t}\epsilon,$$
$$\hat{z} = \Phi_\theta(z_t, 0), \quad \hat{x} = \text{Decoder}(\hat{z}).$$

This definition aligns with the image generation process of the stable diffusion model.

For the Diffusion Transformer, we define the variation API as $V_{\tilde{\theta}}(x, t)$, where $\tilde{\theta}$ corresponds to the Vision Transformer architecture instead of the U-net. We repeatedly call the variation API and calculate the difference between the original image and the reconstructed image, as done in DDIM.

## 5. Experiments

In this section, we evaluate the performance of our methods across various datasets and settings. We follow the same experiment setup in previous papers (Duan et al., 2023; Kong et al., 2023). The detailed experimental settings, including datasets, models, and hyper-parameter settings can be found in Appendix A[1].

### 5.1. Evaluation Metrics

We follow the metrics used in previous papers (Duan et al., 2023; Kong et al., 2023), including Area Under Receiver Operating Characteristic (AUC), Attack Success Rate (ASR), the True Positive Rate (TP) when the False Positive Rate is 1%. We also plot the ROC curves.

### 5.2. MIA with DDIM Models

We follow the experimental setup of (Duan et al., 2023; Kong et al., 2023). We train a DDIM model on the CIFAR-10/100 (Krizhevsky et al., 2009) and STL10-Unlabeled datasets (Coates et al., 2011), using the image generation step $T = 1000$ and sampling interval $k = 100$. For all the datasets, we randomly select 50% of the training samples to train the model and denote them as members. The remain-

---

[1]Code is available at https://github.com/lijingwei0502/diffusion_mia.

Table 2: Comparison of different methods for Diffusion Transformer using the same set of metrics as Table 1. Previous methods require access to the Vision Transformer, whereas our methods do not. We use AUC, ASR, and TP as the metrics, TP refers to the True Positive Rate when the False Positive Rate is 1%.

| Method | | ImageNet $128 \times 128$ | | | ImageNet $256 \times 256$ | | |
|---|---|---|---|---|---|---|---|
| Algorithm | Transformer | AUC | ASR | TP | AUC | ASR | TP |
| Loss (Matsumoto et al., 2023) | □ | 0.83 | 0.76 | 10.7 | 0.78 | 0.70 | 7.3 |
| SecMI (Duan et al., 2023) | □ | 0.80 | 0.73 | 8.3 | 0.88 | 0.80 | 16.3 |
| PIA (Kong et al., 2023) | □ | 0.97 | 0.92 | 32.1 | 0.91 | 0.85 | 6.8 |
| PIAN (Kong et al., 2023) | □ | 0.66 | 0.64 | 6.2 | 0.67 | 0.66 | 12.8 |
| **REDIFFUSE** | ■ | **0.98** | **0.95** | **44.1** | **0.97** | **0.94** | **47.3** |

□: Require the access of Transformer.     ■: Do not require the access of Transformer.

Table 3: Comparison of different methods for Stable Diffusion using the same set of metrics as Table 1. Again, previous methods require access to U-Net, whereas our methods do not. We use AUC, ASR and TP as the metrics, TP refers to the True Positive Rate when the False Positive Rate is 1%.

| Method | | Laion5 | | | Laion5 with BLIP | | |
|---|---|---|---|---|---|---|---|
| Algorithm | U-Net | AUC | ASR | TP | AUC | ASR | TP |
| Loss (Matsumoto et al., 2023) | □ | 0.62 | 0.61 | 13.2 | 0.62 | 0.62 | 13.3 |
| SecMI (Duan et al., 2023) | □ | 0.70 | 0.65 | 19.2 | 0.71 | 0.66 | 19.8 |
| PIA (Kong et al., 2023) | □ | 0.70 | 0.66 | 19.7 | 0.73 | 0.68 | 20.2 |
| PIAN (Kong et al., 2023) | □ | 0.56 | 0.53 | 4.8 | 0.55 | 0.51 | 4.4 |
| **REDIFFUSE** | ■ | **0.81** | **0.75** | **20.6** | **0.82** | **0.75** | **21.7** |

□: Require the access of U-Net.     ■: Do not require the access of U-Net.

ing 50% are utilized as nonmembers. We use (Matsumoto et al., 2023), (Duan et al., 2023), (Kong et al., 2023) as our baseline methods. We fix the diffusion step at $t = 200$, and independently call the variation API 10 times to take the average of the output images as $\hat{x}$. We will discuss the impact of the diffusion step and the average number in Section 5.5.

For the difference function $D(x, \hat{x})$, following the setup in (Duan et al., 2023), we take the pixel-wise absolute value of $x - \hat{x}$ to obtain a difference vector $v$ for each image. Using the ResNet-18 network (He et al., 2016) and denoting it as $f_R$, we perform binary classification on these difference vectors. We use 20% of the data as the training set and obtained the label of each difference vector being classified as a member or nonmember. The difference function is obtained by the negated value of the probability outputed by the neural network predicting as member: $D(x, \hat{x}) = -f_R(v)$.

The result is shown in Table 1. Our method achieves high performance, surpassing several baseline algorithms in most setups, and does not require access to the internal structure of the model. This demonstrates that our algorithm is highly

effective and robust.

### 5.3. MIA with Diffusion Transformers

We train a diffusion transformer model on the ImageNet (Deng et al., 2009) dataset following the setup of (Peebles & Xie, 2023). We randomly select 100,000 images from the ImageNet training set to train the model with resolutions of either $128 \times 128$ or $256 \times 256$. For the membership inference attack setup, 1000 images are randomly chosen from our training set as the member set, and another 1000 images are randomly selected from the ImageNet validation set as the non-member set. We fix the diffusion step at $t = 150$ and the DDIM step at $k = 50$, and we independently call the variation API 10 times to take the average of the output images as $\hat{x}$.

We use (Matsumoto et al., 2023), (Duan et al., 2023), (Kong et al., 2023) as our baseline methods. Since these work did not study the case of Diffusion Transformers, we integrate their algorithms into the DiT framework for evaluation. For the difference function $D(x, \hat{x})$, following the setup in (Duan et al., 2023; Kong et al., 2023), we take the L2 norm of $x - \hat{x}$ to measure the differences between two

image.The results, presented in Table 2, demonstrate that our method outperforms baseline algorithms and does not require access to the Vision Transformer.

We also plot ROC curves for the DDIM train on CIFAR-10 and the Diffusion Transformer train on ImageNet in Appendix B. The curves further demonstrate the effectiveness of our method.

### 5.4. MIA with the Stable Diffusion Model

We conduct experiments on the original Stable Diffusion model, i.e., stable-diffusion-v1-4 provided by Huggingface, without further fine-tuning or modifications. We follow the experiment setup of (Duan et al., 2023; Kong et al., 2023), use the LAION-5B dataset (Schuhmann et al., 2022) as member and COCO2017-val (Lin et al., 2014) as non-member. We randomly select 2500 images in each dataset. We test two scenarios: Knowing the ground truth text, which we denote as Laion5; Not knowing the ground truth text and generating text through BLIP (Li et al., 2022), which we denote as Laion5 with BLIP.

For the difference function $D(x, \hat{x})$, since the images in these datasets better correlate with human visual perception, we directly use the SSIM metric (Wang et al., 2004) to measure the differences between two images. The results, presented in Table 3, demonstrate that our methods achieve high accuracy in this setup, outperforming baseline algorithms by approximately 10%. Notably, our methods do not require access to U-Net.

### 5.5. Ablation Studies

In this section, we alter some experimental parameters to test the robustness of our algorithm. We primarily focus on the ablation study of DDIM and Diffusion Transformer, while the ablation study related to Stable Diffusion is provided in Appendix B.

**The Impact of Average Numbers**   We test the effect of using different averaging numbers $n$ on the results, as shown in Figure 3. It can be observed that averaging the images from multiple independent samples to generate the output $\hat{x}$ further improves accuracy. This observation validates the algorithm design intuition discussed in Section 4.2. Additional figures showing the ASR results are presented in Appendix B.

**The Impact of Diffusion Steps**   We adjust the diffusion step $t$ to examine its impact on the results. The experiments are conducted using the DDIM model on CIFAR-10 with diffusion steps for inference. The outcomes are presented in Figure 4. Our findings indicate that as long as a moderate step is chosen, the attack performance remains excellent, demonstrating that our algorithm is not sensitive to the choice of $t$. This further underscores the robustness of our algorithm. We also plot the change of diffusion step

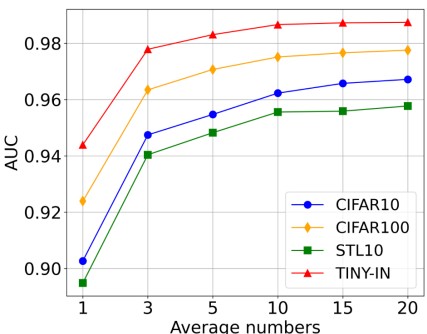

Figure 3: **The impact of average numbers.** We train DDIM model on CIFAR-10. Averaging multiple independent samples proves to be effective in further improving the overall performance of our algorithm, which validates the intuition of our algorithm design.

for other diffusion models and datasets in the Appendix B.

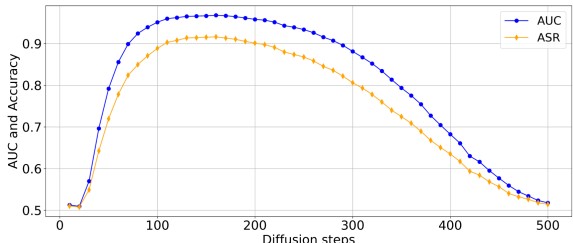

Figure 4: **The impact of diffusion steps on DDIM.** We train a DDIM model on the CIFAR-10 dataset and use different diffusion steps for inference. We find that high accuracy can be achieved as long as a moderate step number is chosen. This opens up possibilities for practical applications in real-world scenarios.

**The Impact of Sampling Intervals**   In DDIM and Diffusion Transformer, the model uses a set of steps denoted by $\tau_1, \tau_2, \ldots, \tau_T$. It samples each of these steps to create the image. The spacing between these steps is referred to as the sampling interval. We change the sampling interval $k$ and check the influence on the results. As shown in Figure 5, we adjust this parameter for attacks on both DDIM and Diffusion Transformer. We find that our method achieves high AUC values across different sampling intervals, demonstrating that our detection capabilities are not significantly limited by this parameter. Additionally, in Appendix B, we plot the effect of different sampling intervals on ASR and find that the impact is minimal.

### 6. An Application to DALL-E's API

In this section, we conduct a small experiment with online API services to test the effectiveness of our algorithm. We test with the DALL-E 2 (Ramesh et al., 2022) model since

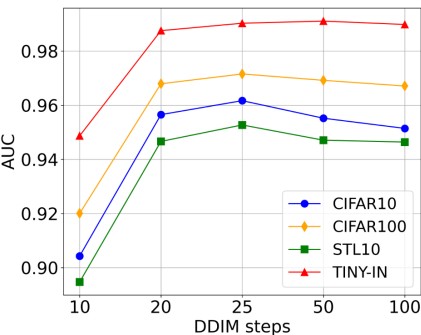

Figure 5: **The impact of sampling intervals.** We train DDIM model on CIFAR-10. We find that adjusting the sampling interval has a relatively small influence in the first case and does not affect our method in the latter case. This makes our algorithm applicable to more setups.

DALL-E 2 provides a variation API service. We select different thresholds and classify an image with a variation error below the threshold as members and those above the threshold as non-members. We then calculate metrics such as AUC and ASR. Since the baseline algorithms (Matsumoto et al., 2023; Duan et al., 2023; Kong et al., 2023) require intermediate results, we are unable to test these algorithms under the online API setup.

One challenge with MIA for DALL-E 2 is that it does not disclose its training set. However, since it is adept at generating various famous artworks, we select 30 famous paintings from five famous artists: Van Gogh, Monet, Da Vinci, Dali, and Rembrandt, to form our member set. We believe that it is reasonable to hypothesize that these artworks are used in DALL-E 2's training set. For constructing the non-member set, we used Stable Diffusion 3 (Esser et al., 2024) to generate images based on the titles of each painting in the member set. The benefit of constructing non-members in this way is that it allows for control over the content of the artwork descriptions, reducing bias caused by content shift. Moreover, these generated images are certainly not in the DALL-E 2's training set.

The results, presented in Table 4, demonstrate that our algorithm achieves a relatively high accuracy under this evaluation method. Our observation is illustrated in Figure 6. As seen in the figure, for Monet's iconic painting "Water Lily Pond", the original artwork shows minimal changes when using DALL-E 2's variation API, retaining most of its main features. In contrast, the artwork generated by Stable Diffusion 3 undergoes significant changes, with variations in both the number of flowers and lily pads. Therefore, we hypothesize that artworks with smaller changes after API usage are more likely to have appeared in the model's training set. Other results of changes to the member and

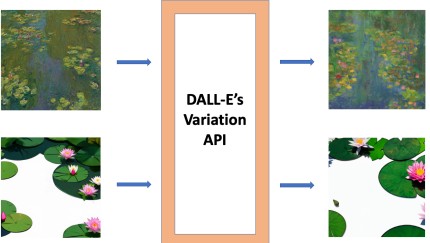

Figure 6: **Main observation of our attack.** DALL-E 2's variation API makes minimal changes to famous artworks, while nonmember images with similar content undergo significant alterations. More examples of changes to the member and non-member inputs after applying the variation API can be found in Appendix D.

non-member inputs after applying the variation API can be found in Appendix D.

Table 4: **The results of applying our algorithm to DALL-E 2's variation API**. We assume some famous paintings as members and use Stable Diffusion 3 along with the titles of these artworks to generate corresponding non-members. Our algorithm also achieves high accuracy under this setup.

| Metrics | $L_1$ distance | $L_2$ distance |
|---------|---------------|---------------|
| AUC     | 76.2          | 88.3          |
| ASR     | 74.5          | 81.4          |

The results indicate that we can apply our algorithm with online API services. We acknowledge that this part of the experimental design has certain limitations. Not every famous painting we selected may be present in DALL-E 2's training set, and our construction of non-members may exhibit some distribution shift relative to the member dataset. Here, we aim to provide a real-world application for black-box evaluation, leaving a more comprehensive experimental design as future work.

## 7. Conclusion, Limitations and Future Directions

In this work, we introduce a novel membership inference attack method specifically designed for diffusion models. Our approach only requires access to the variation API of the model, bypassing the need for internal network components such as the U-net. This represents an advancement for commercial diffusion models, which typically restrict internal access. We demonstrate the effectiveness of our approach across various datasets, showing that it achieves high accuracy in identifying whether an image was included in the training dataset. Our algorithm can detect data misuse

by the model, representing a step forward in protecting the copyright of artworks.

However, our method has certain limitations, particularly the requirement for a moderate diffusion step $t$ in the variation API. The algorithm's accuracy declines when the diffusion step is excessively high. As such, we propose our method as an initial step towards black-box MIA, with a more comprehensive solution left for subsequent exploration.

Future work could focus on developing more robust algorithms capable of handling a broader range of diffusion steps. Improving the interpretability of our method and extending it to other generative models are also valuable directions for further research.

## Impact Statement

This paper addresses the critical issue of protecting intellectual property and data privacy in the era of rapidly advancing generative models. By proposing a novel membership inference attack (MIA) method, our work aims to identify whether specific artworks have been used during the training of diffusion models. This has significant implications for safeguarding creators' copyrights, detecting unauthorized use of artistic works, and promoting ethical practices in the development and deployment of generative models. While our method is designed with privacy protection in mind, we acknowledge the potential for misuse. We hope this research serves as a foundation for further discussions on data privacy and ethical considerations, encouraging responsible use and development of diffusion models in the community.

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

## A. Datasets, Models and Hyperparameters

We use NVIDIA RTX 6000 graphics cards for all our experiments.

For DDIM, we follow the training hyperparameters of (Duan et al., 2023) to train a DDIM model on the CIFAR-10/100 (Krizhevsky et al., 2009) and STL10-Unlabeled (Coates et al., 2011) datasets, using 1000 image generation steps ($T = 1000$) and a sampling interval of $k = 100$. The training iterations are set to 800,000. For all the datasets, we randomly select 50% of the training samples to train the model and designate them as members. The remaining 50% of the training samples are used as nonmembers.

We use (Matsumoto et al., 2023; Duan et al., 2023; Kong et al., 2023) as our baseline methods. We use their official code repositories and apply the optimal hyperparameters from their papers. For our algorithm, we fix the diffusion step at $t = 200$ and independently call the variation API 10 times to average the output images as $\hat{x}$.

For the Diffusion Transformers, we train the model on the ImageNet (Deng et al., 2009) dataset, using 1000 image generation steps ($T = 1000$), while for other training hyperparameters, we follow the setup of (Peebles & Xie, 2023). We randomly select 100,000 images from the ImageNet training set to train the model at resolutions of either $128 \times 128$ or $256 \times 256$. For the $128 \times 128$ image size, we use 200,000 training iterations. For the $256 \times 256$ image size, we use 300,000 training iterations. These numbers of training iterations are chosen to ensure the generation of high-quality images.

For the membership inference attack setup, 1000 images are randomly selected from our training set as the member set, and another 1000 images are randomly chosen from the ImageNet validation set as the non-member set. We fix the diffusion step at $t = 100$ and the DDIM step at $k = 50$, and independently call the variation API 10 times to average the output images as $\hat{x}$.

We also use (Matsumoto et al., 2023; Duan et al., 2023; Kong et al., 2023) as our baseline methods. Since these papers do not study Diffusion Transformers, we adapt their algorithms to the DiT framework for evaluation. We fix the DDIM step at $k = 50$ and choose diffusion steps $t \in [50, 100, 150, 200, 250, 300]$, recording their optimal solutions under these different hyperparameter settings.

For the Stable Diffusion experiments, we use the original Stable Diffusion model, i.e., stable-diffusion-v1-4 provided by Huggingface, without further fine-tuning or modifications. We follow the experimental setup of (Duan et al., 2023; Kong et al., 2023), using an image generation step of $T = 1000$ and a sampling interval of $k = 10$. We use the LAION-5B dataset (Schuhmann et al., 2022) as the member set and COCO2017-val (Lin et al., 2014) as the non-member set. We randomly select 2500 images from each dataset. We test two scenarios: knowing the ground truth text, denoted as Laion; and generating text through BLIP (Li et al., 2022), denoted as Laion with BLIP.

We use the hyperparameters from the papers (Duan et al., 2023; Kong et al., 2023) to run the baseline methods. For our algorithms REDIFFUSE, we fix the diffusion step at $t = 10$ to call the variation API and directly use the SSIM metric (Wang et al., 2004) to measure the differences between two images. Other hyperparameters remain the same as in the baseline methods.

# B. More Experiment Results

In this section, we show other experiment results which is not in our main paper. We conduct more ablation studies.

**The Impact of Average Numbers**  We use different average numbers $n$ and test the influence on the results. Besides the figures of AUC in the main paper, the figures of ASR of DDIM and Diffusion Transformer are also plotted in Figure 7. In addition, we plot the figures of AUC and ASR for Stable Diffusion in Figure 8. We observe that in the DDIM and Diffusion Transformer setup, averaging the images from multiple independent samples as the output further improves accuracy. In the stable diffusion setup, since the image size in the dataset is larger (512x512), the reconstructed images are more stable and not influenced by perturbations at specific coordinates. Therefore, averaging multiple images is not necessary.

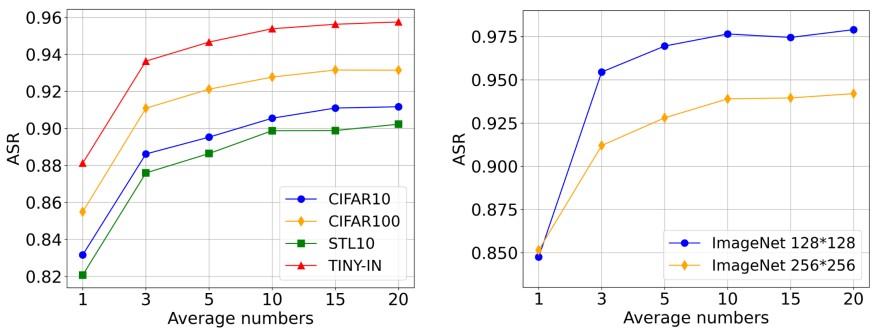

Figure 7: **The impact of average numbers.** Left: DDIM model on CIFAR-10. Right: Diffusion Transformer model on Imagenet. Averaging can further improve the performance of our algorithm.

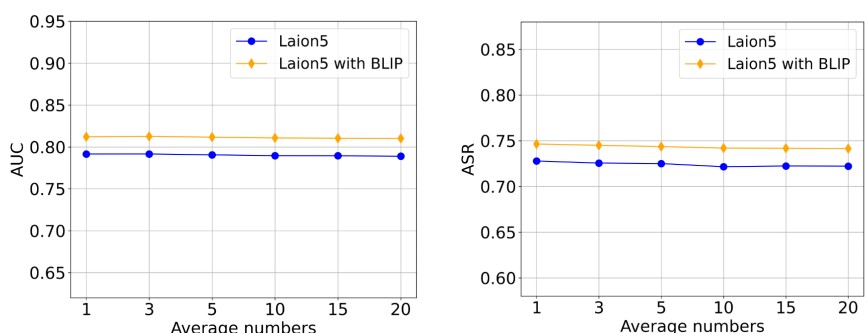

Figure 8: **The impact of average numbers on Stable Diffusion.** We plot the AUC and ASR metrics, and averaging does not improve performance.

**The Impact of Diffusion Steps**  We adjust the diffusion step $t$ to examine its impact on the results. We train the DDIM model on CIFAR100 (Figure 9), STL10 (Figure 10) dataset, Diffusion Transformer on the Imagenet $256 \times 256$ (Figure 11)dataset and Stable Diffusion on Laion5 dataset (Figure 12). From the results, we see that our algorithm can achieve high performance over a wide range of diffusion steps. This opens up possibilities for practical applications in real-world scenarios.

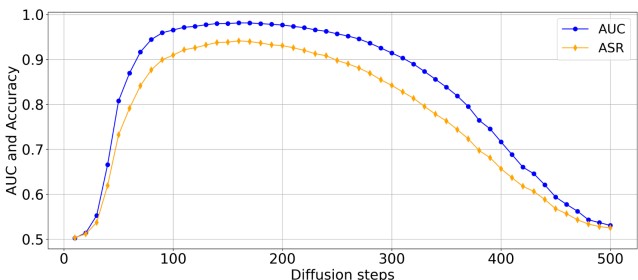

Figure 9: **The impact of diffusion steps.** We train a DDIM model on the CIFAR-100 dataset and use different diffusion step for inference.

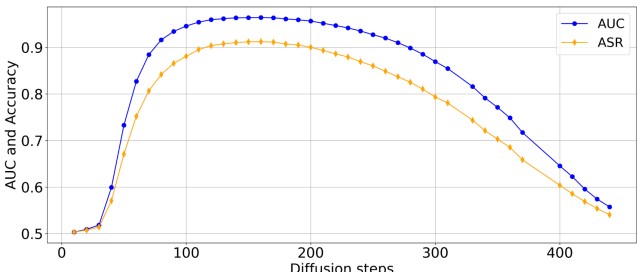

Figure 10: **The impact of diffusion steps.** We train a DDIM model on the STL-10 dataset and use different diffusion step for inference.

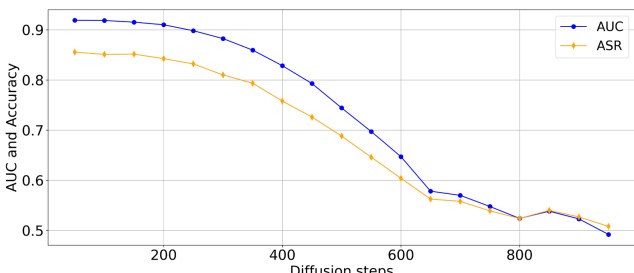

Figure 11: **The impact of diffusion steps.** We train a Diffusion Transformer model on the Imagenet $256 \times 256$ dataset and use different diffusion step for inference. The robust results imply that our algorithm is also not very sensitive to the choice of $t$ in this setup.

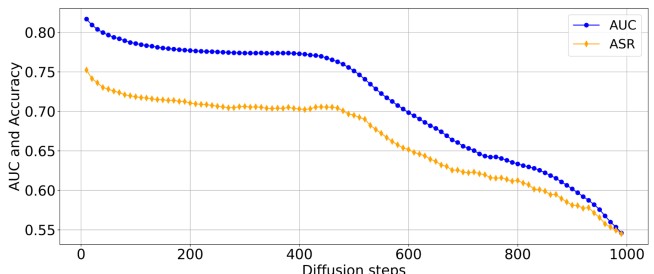

Figure 12: **The impact of diffusion steps.** We use the Stable Diffusion model with the Laion5 dataset for evaluation and test different diffusion steps for inference. Specifically, using relatively small diffusion steps results in better performance.

**The Impact of Sampling Intervals**    We change the sampling intervals to see if there is any influence on the results. In addition to the AUC figures in the main paper, the ASR figures of DDIM and Diffusion Transformer are also plotted in Figure 13. We also plot the AUC and ASR for Stable Diffusion in Figure 14. From the results, we observe that our algorithm consistently performs well across different sampling intervals.

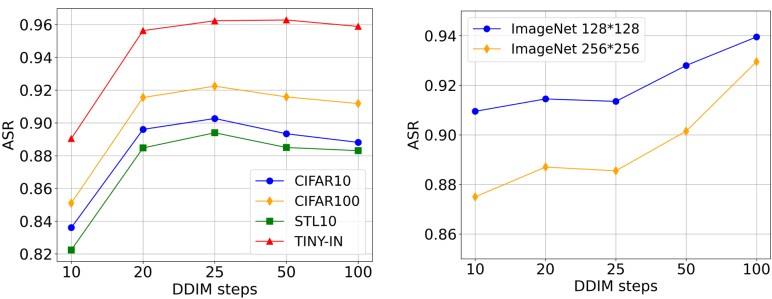

Figure 13: **The impact of sampling intervals.** Left: DDIM model on CIFAR-10. Right: Diffusion Transformer on Imagenet. We find that adjusting the sampling interval does not significantly affect of the accuracy.

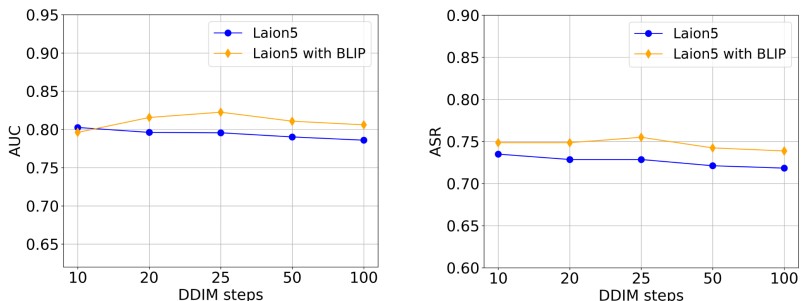

Figure 14: **The impact of sampling intervals on Stable Diffusion.** We plot the AUC and ASR metrics, observe that different sampling intervals have minimal impact on it.

**The ROC curves**    We plot ROC curves for the DDIM train on CIFAR-10 and Diffusion Transformer train on ImageNet $256 \times 256$ in Figure 15. The curves further demonstrate the effectiveness of our method.

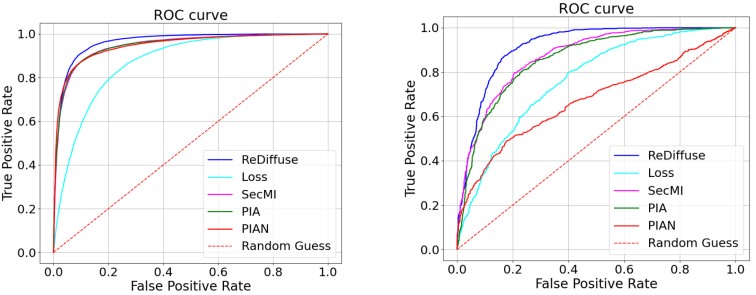

Figure 15: **The ROC curves of various setups.** Left: DDIM model on CIFAR-10. Right: Diffusion Transformer on ImageNet $256 \times 256$. The curves show that our algorithm outperforms the baseline algorithms.

## C. Proof

In this section, we present the proof of the theorem.

**Theorem 4.2.** *Suppose the DDIM model can learn a parameter $\theta$ such that, for any $x \sim D_{training}$ with dimension $d$, the prediction error $\epsilon - \epsilon_\theta(\sqrt{\bar{\alpha}_t}x + \sqrt{1 - \bar{\alpha}_t}\epsilon, t)$ is a random variable $X = (X_1, X_2, \ldots, X_d)$ with zero expectation and finite cumulant-generating function for each coordinate (Durret, 2010). Suppose the sampling interval $k$ is equal to the variation API diffusion step $t$. Let $\hat{x}$ be the average of $n$ output images of $x$ using the variation API. Then we have*

$$\mathbb{P}(\|\hat{x} - x\| \geq \beta) \leq d \exp\left(-n \min_i \Psi^*_{X_i}\left(\frac{\beta\sqrt{\bar{\alpha}_t}}{\sqrt{d(1 - \bar{\alpha}_t)}}\right)\right),$$

*where $\beta > 0$ and $\Psi^*_{X_i}$ is the Fenchel-Legendre dual of the cumulant-generating function $\Psi_{X_i}$.*

*Proof.* We denote $x^i$ as the $i$-th output image of $x$ using the variation API. We denote the $i$-th Gaussian noise as $\epsilon^i$ and the forward process yields $x^i_t = \sqrt{\bar{\alpha}_t}x + \sqrt{1 - \bar{\alpha}_t}\epsilon^i$.

As the sampling interval is equal to the variation API diffusion step $t$, we have

$$\begin{aligned}
x^i - x &= \frac{x^i_t - \sqrt{1 - \bar{\alpha}_t}\epsilon_\theta(\sqrt{\bar{\alpha}_t}x + \sqrt{1 - \bar{\alpha}_t}\epsilon^i, t)}{\sqrt{\bar{\alpha}_t}} - x, \\
&= \frac{\sqrt{\bar{\alpha}_t}x + \sqrt{1 - \bar{\alpha}_t}\epsilon^i - \sqrt{1 - \bar{\alpha}_t}\epsilon_\theta(\sqrt{\bar{\alpha}_t}x + \sqrt{1 - \bar{\alpha}_t}\epsilon^i, t)}{\sqrt{\bar{\alpha}_t}} - x, \\
&= \frac{\sqrt{1 - \bar{\alpha}_t}}{\sqrt{\bar{\alpha}_t}}(\epsilon^i - \epsilon_\theta(\sqrt{\bar{\alpha}_t}x + \sqrt{1 - \bar{\alpha}_t}\epsilon^i, t)).
\end{aligned}$$

If we denote the random variable $X^i = (X^i_1, X^i_2, \ldots, X^i_d)$ represents $\epsilon^i - \epsilon_\theta(\sqrt{\bar{\alpha}_t}x + \sqrt{1 - \bar{\alpha}_t}\epsilon^i, t)$, then we consider $S^i_n = \sum_{j=1}^n X^i_j$. From the assumption we know that all the $X^i_j$ are i.i.d. random variables(with the same distribution of $X_i$ in Theorem 4.2) with zero expectation and finite cumulant-generating function $\Psi_{X_i}(s) = \log E[e^{sX_i}] < +\infty$. Using the theorem of Cramér–Chernoff method for sums of i.i.d. random variables (Durret, 2010), we get the following probability inequality for any $\beta > 0$:

$$\mathbb{P}(|S^i_n| \geq \beta) \leq \exp(-n\Psi^*_{X_i}(\frac{\beta}{n})),$$

where $\beta > 0$ and $\Psi^*_{X_i}(y) = \sup_{s>0}(sy - \Psi_{X_i}(s))$ is the Fenchel-Legendre dual of the cumulant-generating function $\Psi_{X_i}$.

Therefore, denote $S_n = (S^1_n, S^2_n, \ldots, S^d_n)$, taking the definition of $\|\hat{x} - x\| = \|\frac{1}{n}\sum_{i=1}^n(x^i - x)\| = \frac{\sqrt{1 - \bar{\alpha}_t}}{n\sqrt{\bar{\alpha}_t}}\|S_n\|$, we get the following bound of the reconstruction error:

$$\mathbb{P}(\|\hat{x} - x\| \geq \beta) \leq \sum_{i=1}^d P(|S^i_n| \geq \frac{n\beta\sqrt{\bar{\alpha}_t}}{\sqrt{d(1 - \bar{\alpha}_t)}}) \leq d \exp(-n \min_i \Psi^*_{X_i}(\frac{\beta\sqrt{\bar{\alpha}_t}}{\sqrt{d(1 - \bar{\alpha}_t)}})),$$

So averaging the randomly reconstructed data from a training set will result in a smaller reconstruction error with high probability of $1 - \mathcal{O}(\exp(-n))$ when we use a large $n$.

$\square$

# D. More Results of Variation Images

In this section, we provide more results of the variation of member and nonmember image when applying to DALL-E 2's variation API. The experimental results are recorded in Figure 16, from which we can see that the variation in the member inputs after applying the DALL-E 2's variation API is relatively small than non-member inputs.

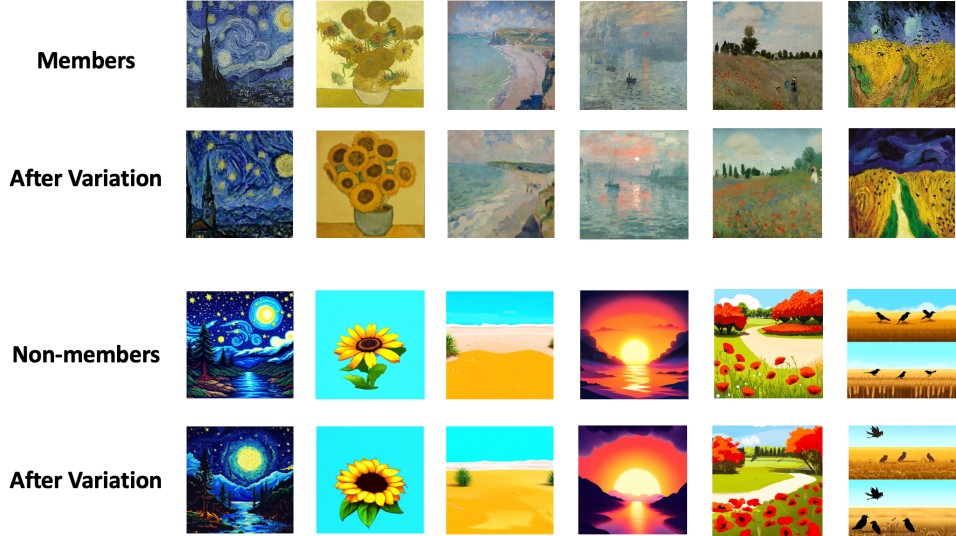

Figure 16: **More results of the variation images.** From the figures we can see that DALL-E 2's variation API makes less changes to images in member set than non-member set.

