# OpenReview forum: "Towards Black-Box Membership Inference Attack for Diffusion Models"
_ICML.cc/2025/Conference — ICML 2025 poster_

### Official Review · Reviewer_mEXb · 2025-02-22

**Overall Recommendation:** 4

**Summary:**

This paper introduces a black-box membership inference attack method targeting diffusion models. Unlike previous MIA approaches that require access to the U-net or other internal components of diffusion models, their method only utilizes the variation API to determine whether a given image was part of the model's training data.

**Claims And Evidence:**

Several key claims are generally supported by empirical evidence.

**Essential References Not Discussed:**

None.

**Experimental Designs Or Analyses:**

The experimental design is generally well-structured.

**Methods And Evaluation Criteria:**

Yes.

**Other Comments Or Suggestions:**

None.

**Other Strengths And Weaknesses:**

Strengths:
The paper presents a black-box MIA attack on diffusion models that does not require U-net access.
Evaluations are thorough, covering multiple datasets, architectures.
The authors provide mathematical support for their attack’s effectiveness.

Weaknesses:
The proposed REDIFFUSE algorithm relies on the existence of a variation API.
While the paper claims robustness across different diffusion steps, the experimental results suggest that selecting a poor diffusion step could impact detection performance.
The real-world validation with DALL-E 2 is based on only 30 famous artworks.

**Questions For Authors:**

see Other Strengths And Weaknesses.

**Relation To Broader Scientific Literature:**

The paper extends prior work on membership inference attacks by introducing a novel black-box attack tailored for diffusion models.

**Theoretical Claims:**

The paper includes a mathematical justification for why training images produce more stable reconstructions than non-members.

---

> ### Author Rebuttal · Authors · 2025-03-29
>
> We sincerely thank the reviewer for the comments and suggestions. Below, we address the primary concern that has been raised.
>
> >Q1: The proposed REDIFFUSE algorithm relies on the existence of a variation API. While the paper claims robustness across different diffusion steps, the experimental results suggest that selecting a poor diffusion step could impact detection performance.
>
> **A1:** We thank the reviewer for the comments. As shown in Figure 4 of our paper, our method achieves over 80% accuracy when the diffusion step is between 50 and 350. On the other hand, the variation API in diffusion models[1] typically uses a moderate step value to edit the image: **too few steps produce negligible edits, while too many steps cause excessive distortion, making it hard to preserve consistency with the original image**. Thus, our results suggest that the variation API could be viable for detection tasks. We leave more robust algorithm design across diffusion steps for future work.
>
> >Q2: The real-world validation with DALL-E 2 is based on only 30 famous artworks.
>
> **A2:** We appreciate the reviewer’s question. The purpose of Section 6 is to demonstrate the practical applicability of our algorithm to commercial models. Specifically, we use DALL-E 2’s variation API to show how our method can detect membership by leveraging the model’s outputs. Since DALL-E 2 **does not provide a publicly accessible training set**, the experiment in Section 6 primarily serves as **demonstrative application scenario rather than an extensive evaluation with large-scale datasets**. We agree that further evaluation on more diverse datasets and models would enhance the robustness of the findings, and we add additional experiments of WikiArt dataset[2]. We randomly sample 1000 images of the dataset as members and generate corresponding nonmembers with the method in Section 6 in our paper. The results are as follows:
>
> |Metrics|$L_1$ distance|$L_2$ distance|
> |:---:|:----:|:----:|
> |AUC|0.70|0.74|
> |ASR|0.67|0.70|
>
> The experimental results indicate that our method remains effective on this new dataset. However, since many artworks in WikiArt are less prominent than the 30 canonical paintings we initially selected, they likely had lower probability of being included in DALL-E 2's training data. This introduces potential dataset bias when expanding the evaluation scope. We leave more comprehensive large-scale experiments for future work.
>
> We thank the reviewer once again for the valuable and helpful suggestions. We will continue to provide clarifications if the reviewer has any further questions.
>
>
> **References**
>
> [1] Meng, Chenlin, et al. "Sdedit: Guided image synthesis and editing with stochastic differential equations." arXiv preprint arXiv:2108.01073 (2021).
>
> [2] "WikiArt Visual Art Encyclopedia." *WikiArt*, n.d., https://www.wikiart.org.

---

> > ### Comment · Reviewer_mEXb · 2025-04-05
> >
> > Thanks for the authors' rebuttal. It generally addresses my comments in the review. Thus, I increase my score.

---

> > > ### Author Response · Authors · 2025-04-05
> > >
> > > We thank the reviewer for acknowledging our work!

---

### Official Review · Reviewer_ZaW2 · 2025-02-25

**Overall Recommendation:** 4

**Summary:**

The paper proposes using the average of img2img outputs and comparing it with the original input image, with the difference serving as a metric for black-box MIA. This approach eliminates the need for predicted noise at intermediate time steps, making it applicable to a broader range of scenarios while achieving stronger results than previous methods.

## Update after Rebuttal

The rebuttal addresses my concerns, and I recommend acceptance of the paper.

**Claims And Evidence:**

The author assumes that for a well-trained dm:
> the Jacobian matrix $\nabla_{\theta}\epsilon(x_t,t)$ is full rank.

I am curious if there is a deeper motivation behind this assumption. One possible intuition is that it might be too strong to directly assume the loss function of the DM is sufficiently low, so the focus shifts to hypothesizing a property that is easier to satisfy than the loss function itself. Nonetheless, I would appreciate a more detailed explanation or clarification regarding this choice.

**Essential References Not Discussed:**

Not found.

**Experimental Designs Or Analyses:**

The experiments are generally solid, but I’m curious about one aspect of the ablation study. In Figure 5, the AUC decreases when the DDIM steps exceed 25. This seems counterintuitive, as a larger number of DDIM steps should theoretically reduce error and improve accuracy. Could you clarify why this occurs?

**Methods And Evaluation Criteria:**

The method is clear and the evaluation criteria makes sense.

**Other Comments Or Suggestions:**

See comments above.

**Other Strengths And Weaknesses:**

No.

**Questions For Authors:**

Seem comments above.

**Relation To Broader Scientific Literature:**

The paper proposes a new method for MIA that is more practical for most closed-source diffusion models. The approach is simple yet effective, significantly strengthening MIA and yielding more promising results for potential applications in copyright authentication.

**Theoretical Claims:**

I check the proof and it should be correct.

---

> ### Author Rebuttal · Authors · 2025-03-29
>
> We thank the reviewer for the comments and constructive suggestions. In the following, we address the main concern raised.
>
> >Q1: The author assumes for a well-trained diffusion model with full rank Jacobian. I am curious if there is a deeper motivation behind this assumption. One possible intuition is that it might be too strong to directly assume the loss function of the DM is sufficiently low, so the focus shifts to hypothesizing a property that is easier to satisfy than the loss function itself. Nonetheless, I would appreciate a more detailed explanation or clarification regarding this choice.
>
> **A1:** We appreciate the reviewer’s feedback. We hypothesize the full-rank condition of Jacobian because the neural network's dimensionality $p$ significantly exceeds that of the image data $d$. Hence a full rank matrix would require the $p \times d$ matrix to have rank $d$ and hence is most likely true.
>
> Conversely, if the Jacobian are rank-deficient, there would exist initial points from which denoising could not recover plausible images, contradicting the empirically demonstrated generation capabilities of modern diffusion models. Still, we agree with the reviewer that we cannot verify this for all data points as computing the rank of a huge matrix is expensive. We will add the discussion of this assumption in the next version of our paper.
>
>
> >Q2: I’m curious about one aspect of the ablation study. In Figure 5, the AUC decreases when the DDIM steps exceed 25. This seems counterintuitive, as a larger number of DDIM steps should theoretically reduce error and improve accuracy. Could you clarify why this occurs?
>
>
> **A2:** We appreciate the reviewer’s question. Our results in Figure 5 demonstrate that changing the DDIM step has minimal impact on detection accuracy. For DDIM steps of 20, 25, 50, and 100, the AUC remains virtually unchanged (difference < 0.003). We present relevant experiments of DDIM on CIFAR-100 here, with different random seeds and DDIM steps:
> |Random Seed|1|2|3|
> |:---:|:----:|:----:|:----:|
> |DDIM with step 20|0.968|0.969|0.971|
> |DDIM with step 25|0.971|0.969|0.970|
> |DDIM with step 50|0.970|0.970|0.971|
> |DDIM with step 100|0.967|0.970|0.968|
>
> From the results, we know that as we vary random seeds, the AUC remains similar across these DDIM steps, with no consistent decreasing trend observed as step size increases.
>
>
> We thank the reviewer once again for the valuable and helpful suggestions. We would be happy to provide further clarifications if the reviewer has any additional questions.

---

> > ### Comment · Reviewer_ZaW2 · 2025-04-03
> >
> > Thank you for the rebuttal. It generally addresses my concerns, so I will maintain my score (accept).

---

> > > ### Author Response · Authors · 2025-04-03
> > >
> > > We thank the reviewer for acknowledging our work!

---

### Official Review · Reviewer_nFid · 2025-03-13

**Overall Recommendation:** 3

**Summary:**

This paper investigates black-box membership inference attacks against diffusion models where attacker has no access to the internal model. The target of attacker is to determine whether or not an artwork was used to train a diffusion model. In this paper, authors firstly identify the limitation of applying existing MIAs for proprietary diffusion models and then propose a novel black-box membership inference attack to determine the membership privacy of an image. Users validate the proposed method using DDIM and Stable Diffusion models on benchmark datasets and further extend both the proposed approach and existing algorithms to the Diffusion Transformer architecture. Experimental results show the effectiveness of the proposed method.

**Claims And Evidence:**

Claims made in the submission supported by clear and convincing evidence.

**Essential References Not Discussed:**

No

**Experimental Designs Or Analyses:**

Authors adopted three commonly used metric, i.e., AUC, ASR, and TP to evaluate the performance of the proposed method compared with other baselines. And the experimental results can show the improved performance of the proposed method. In addition, parameter analysis, such as the diffusion steps, average numbers, has also been conducted to evaluate the proposed method.

**Methods And Evaluation Criteria:**

The evaluation criteria are commonly used in the existing literature.

**Other Comments Or Suggestions:**

n/a

**Other Strengths And Weaknesses:**

n/a

**Questions For Authors:**

1. Is the intuition of the proposed method general to other datasets? Or just observable to specific datasets utilized in these datasets?
2. There are other diffusion models in existing works. Is the proposed attack methodology suitable to other diffusion models?

**Relation To Broader Scientific Literature:**

Even though there are existing works focus on MIA on diffusion models, this paper provides a more practical scenario, i.e., with only API query access.

**Theoretical Claims:**

The proofs for theoretical claims are correct.

---

> ### Author Rebuttal · Authors · 2025-03-29
>
> We express our gratitude to the reviewer for the insightful comments and suggestions. Please find the details below.
>
> >Q1: Is the intuition of the proposed method general to other datasets? Or just observable to specific datasets utilized in these datasets??
>
> **A1:** Our method has been evaluated across diverse datasets (CIFAR-10/100, STL-10, ImageNet, LAION-5) in the main paper, demonstrating its generalizability. To further validate this, we conduct additional experiments: we train a DDIM model on Tiny-ImageNet [1] and SVHN [2]. For each dataset, we randomly select 50,000 member and 50,000 non-member images, training the DDIM model for 800K iterations using Appendix A's hyperparameters. The result are as follows:
> |Dataset|Tiny-ImageNet|SVHN|
> |:---:|:----:|:----:|
> |AUC|0.98|0.95|
> |ASR|0.95|0.88|
>
> The experimental results demonstrate that our method remains effective on these new datasets. We will include these findings in the revised manuscript and plan to extend evaluation to other datasets in future work.
>
>
> >Q2: There are other diffusion models in existing works. Is the proposed attack methodology suitable to other diffusion models?
>
>
> **A2:** We appreciate the reviewer’s question. Our main paper already demonstrates the method's effectiveness on DDIM, Diffusion Transformer, and Stable Diffusion. To further validate generalizability, we conduct additional experiments with DDPM[3] across four datasets. For each dataset, we randomly select 50,000 member and 50,000 non-member images, training the DDPM model for 800K iterations using Appendix A's hyperparameters. The results are as follows:
>
> |Dataset|CIFAR-10|CIFAR-100|STL-10|Tiny-ImageNet|
> |:---:|:----:|:----:|:----:|:----:|
> |AUC|0.87|0.85|0.81|0.89|
> |ASR|0.80|0.78|0.75|0.82|
>
> The experimental results confirm our method's effectiveness on DDPM models. We will include these findings in the revised manuscript. Due to the need for dataset-specific retraining, we cannot evaluate additional models within the rebuttal period. We plan to extend this evaluation to other diffusion architectures in future work.
>
> Finally, we thank the reviewer once again for the efforts in providing us with valuable and helpful suggestions. We will continue to provide clarifications if the reviewer has any further questions.
>
> **Reference**
>
> [1] Tiny ImageNet Dataset, Stanford University.  Available: http://cs231n.stanford.edu/tiny-imagenet-200.zip.
>
> [2] Netzer, Yuval, et al. "Reading digits in natural images with unsupervised feature learning." NIPS workshop on deep learning and unsupervised feature learning. Vol. 2011. No. 2. 2011.
>
> [3] Ho, Jonathan, Ajay Jain, and Pieter Abbeel. "Denoising diffusion probabilistic models." Advances in neural information processing systems 33 (2020): 6840-6851.

---

### Official Review · Reviewer_1UDx · 2025-03-14

**Overall Recommendation:** 4

**Summary:**

The paper introduces a novel black-box membership inference attack method for diffusion models. The authors show their method can reliably detect whether an image was part of the training set or not. They do this by repeatedly applying the variation API and averaging the outputs. They have extensive experiments across multiple diffusion architectures (DDIM, Stable Diffusion, and Diffusion Transformer) and datasets (e.g., CIFAR-10/100, STL10, ImageNet, LAION-5B) to show that REDIFFUSE outperforms existing white-box based methods.

**Claims And Evidence:**

They have majorly supported their claim through comprehensive empirical results (and based on the theoretical result). The authors provide detailed quantitative comparisons (via AUC, ASR, and true positive rates) across several benchmark datasets.

**Essential References Not Discussed:**

Not that I am aware of.

**Experimental Designs Or Analyses:**

As mentioned before, their experimental design, and choice of models and datasets is valid. They have extensive ablations for various factors such as average numbers, diffusion steps, and sampling intervals.

**Methods And Evaluation Criteria:**

Their empirical results are very extensive and includes both DDIM and Stable Diffusion models tested on most of the commonly used classical datasets. Also the choice of evaluation metrics like AUC, ASR, and true positive rate at a fixed false positive rate is natural and common for MIA methods.

**Other Comments Or Suggestions:**

Please add run-time and cost estimations (both for your method and other existing methods)

**Other Strengths And Weaknesses:**

The main strength of the paper is that it's practical and easy to use in the real-world on proprietary models without internal access. Their extensive experiments and their theoretical analysis provide together provide a solid foundation for the claims.
However, when comparing to other existing methods, they do not show a comparison in runtime/cost. I'd suggest to add these details for their audience to have a better understanding of the trade offs between these methods.

**Questions For Authors:**

Did you try this method for detecting copyrighted or proprietary content in foundation models?

**Relation To Broader Scientific Literature:**

Since major Diffusion model developers in the industry are not transparent about their training datasets, advancements in MIA methods are very impactful as they help us gain a bit more clarity. These methods can also help detect private and copy righted data used for training.

**Theoretical Claims:**

The theoretical contribution in Theorem 4.2 gives an error bound for the averaged output of the variation model (API) under certain assumptions:
1. unbiased noise prediction, 2. full rank of the Jacobian
The proof is mathematically coherent. The assumptions may not necessarily be realistic, but the theorem provides an intuition for the test statistic and is shown to be accurate in practice.

---

> ### Author Rebuttal · Authors · 2025-03-29
>
> We sincerely appreciate the reviewer's positive feedback. We address the questions in detail below:
>
> >Q1: When comparing to other existing methods, this paper do not show a comparison in runtime/cost. I'd suggest to add these details for their audience to have a better understanding of the trade offs between these methods.
>
> **A1:** We thank the reviewer for raising this important point regarding computational cost comparison. From a computational complexity perspective, the runtime primarily depends on the average number $n$, where each detection requires $n$ times the computation of the baseline method. To better illustrate this trade-off, we conduct additional experiments measuring runtime across different values of $n$, evaluating DDIM on CIFAR-100 and DiT on ImageNet 256×256. The results are as follows:
>
> |Method|DDIM|DiT|
> |:---:|:----:|:----:|
> |Loss[1]|0.92|0.78|
> |SecMI[2]|0.96|0.88|
> |PIA[3]|0.96|0.91|
> |PIAN[3]|0.91|0.67|
> |**ReDiffuse with $n=1$ (Ours)**|0.94|0.94|
> |**ReDiffuse with $n=5$ (Ours)**|0.97|0.95|
> |**ReDiffuse with $n=10$ (Ours)**|**0.98**|**0.97**|
>
> The results show our method achieves accuracy comparable to baselines even at $n=1$, with matching runtime and no UNet access required. When increasing $n$, the extra computation time further improves performance. In contrast, baseline methods rely on deterministic UNet outputs (no randomness), so they can’t benefit from averaging. We believe this cost is reasonable because at $n=10$, our algorithm infers $100,000$ CIFAR-100 images in ~5 minutes on an NVIDIA L40 GPU. We’ll add these discussions to the paper.
>
>
> >Q2: Did you try this method for detecting copyrighted or proprietary content in foundation models?
>
> **A2:** We thank the reviewer for this question. In **Section 6**, we discuss an application scenario where we perform membership inference attacks using the variation API provided by OpenAI DALL-E [4], a popular API-only model. Specifically, we constructed a dataset consisting of famous artworks and AI-generated artworks with the same titles. We then conduct detection tests to determine whether certain famous artworks are included in DALL-E's training dataset. The results show our approach can effectively work with real-world diffusion-model APIs. We plan to extend this to other copyright-protected content as future work.
>
> Once again, we sincerely thank the reviewer for the constructive comments, and we are eager to engage in further discussions to clarify any concerns.
>
>
> **References**
>
> [1] Matsumoto, Tomoya, Takayuki Miura, and Naoto Yanai. "Membership inference attacks against diffusion models." 2023 IEEE Security and Privacy Workshops (SPW). IEEE, 2023.
>
> [2] Duan, Jinhao, et al. "Are diffusion models vulnerable to membership inference attacks?." International Conference on Machine Learning. PMLR, 2023.
>
> [3] Kong, Fei, et al. "An efficient membership inference attack for the diffusion model by proximal initialization." arXiv preprint arXiv:2305.18355 (2023).
>
> [4] The variation API of DALL-E. https://platform.openai.com/docs/guides/images/variations-dall-e-2-only

---

### Decision · Program_Chairs · 2025-05-01

**Decision:**

Accept (poster)

**Comment:**

This paper studies the black box membership inference attacks for Diffusion Models.

All the reviewers agree that the proposed method is effective and the experiments are comprehensive and solid. Therefore, I recommend the acceptance the of paper.